# New Resident Training Strategy Based on Gamification Techniques: An Escape Room on Sepsis in Children

**DOI:** 10.3390/children9101503

**Published:** 2022-09-30

**Authors:** Carme Alejandre, Patricia Corniero, Gemma Claret, Carlos Alaez, Elisabeth Esteban, Iolanda Jordan

**Affiliations:** 1Paediatric Intensive Care Unit (PICU), Institut de Recerca H. Sant Joan de Déu, Hospital Sant Joan de Déu, Esplugues de Llobregatb, 08950 Barcelona, Spain; 2Paediatric Emergency Service, Institut de Recerca H. Sant Joan de Déu, Hospital Sant Joan de Déu, Esplugues de Llobregat, 08950 Barcelona, Spain; 3Hospital Sant Joan de Déu Barcelona Simulation Programme, Esplugues de Llobregat, 08950 Barcelona, Spain; 4Centro de Investigación Biomédica En Red en Enfermedades Respiratorias, Universitat de Barcelona, 08007 Barcelona, Spain

**Keywords:** gamification, escape room, game-based learning, simulation, paediatrics

## Abstract

Aim: Adapting “escape rooms” for educational purposes is an innovative teaching method. The aim of this study was to ascertain the degree of learning of the residents. A secondary objective was to determine their degree of satisfaction. Methods: A prospective, observational study took place in October 2019. A sepsis-based escape room was designed and carried out. A mix of paediatric medical residents and paediatric nursing residents were enrolled. A prior knowledge test was carried out, which was repeated right at the end of the escape room and then again three months later. Furthermore, all participants completed an anonymous post-study survey. Results: We enrolled 48 residents, 79.2% of whom were women. The mean score for the pre-escape room exam was 7.85/9 (SD 1.65), that for the post-escape room exam was 8.75/9 (SD 0.53), and for the exam three months later, it was 8.30/9 (SD 0.94). Among the participants, 18.8% did not manage to leave before the established 60 min time limit. The results of the satisfaction survey showed high participant satisfaction. Conclusions: The escape room proved to be a valuable educational game that increased students’ knowledge of sepsis management and showed a positive overall perceived value among the participants.

## 1. Background

Traditionally, educational techniques have been based on lecture-type classes given by teaching staff. In recent years, however, several teaching methods have been proposed that focus on improving learning, in which students become active participants, ceasing to be passive spectators in their training. Trends in global health education are increasingly moving toward a diversification of strategies to improve knowledge and skills [1].

Gamification consists of applying strategies based on games in typically serious environments in order to enhance motivation, convey a message, and teach content through the active participation of the “players” [2,3,4]. This system has attracted the interest of educators and researchers due to its ability to enhance transversal skills such as teamwork, leadership, creative thinking, and communication [5,6].

One of these new gamification techniques is the escape room. An escape room is a physical and mental game experience in which participants are challenged to leave a room in which they are trapped. To do this, they must overcome a series of challenges of different types and follow a story that will lead them to the key that opens the exit door, all before the time allotted ends.

Escape rooms were first used in Japan in 2007 [7], and their popularity rapidly grew beginning in 2012–2013, expanding first throughout Asia (starting in Singapore), followed by Europe (starting in Hungary), and then in Australia and North America. It is not clear what the precursor of these activities was, but it was probably a combination of different activities with common elements, such as treasure hunts, point-and-click adventure games, or even adventure movies [8].

An educational escape room is based on the game-based learning that takes place in a real escape room in order to promote knowledge acquisition, teamwork, and communication among the participants. Several studies have portrayed the successful use of educational escape rooms in a wide variety of disciplines ranging from health care and pharmaceutical practice to telecommunications and mathematics, among others [9,10,11,12,13,14,15,16,17,18,19,20,21,22,23,24,25,26,27,28,29,30,31,32,33,34,35,36,37].

With the idea of trying out a different form of training for the resident doctors and nurses at our hospital, we used an escape room to develop a clinical case of a paediatric patient affected by sepsis (Appendix A). The aim of this study was to find out how much residents would learn and to what degree they would develop leadership and teamwork strategies in a stressful environment and with a limited time. A secondary objective was to determine their degree of satisfaction with this new learning technique.

## 2. Material and Methods

A prospective and observational study was designed. The paediatric medical and nursing residents at Hospital Sant Joan de Déu (Barcelona, Spain) who voluntarily signed the informed consent were included. The escape room was held in our hospital’s simulation room, which is equipped with cameras and microphones, in October 2019. There were no exclusion criteria.

Two game masters specializing in critical child care and with extensive experience in teaching and simulation prepared an escape room based on a real case of a paediatric patient with sepsis who was in an emergency room. Nicholson’s RECIPE mnemonic (reflection, engagement, choice, information, play, exposition) for meaningful gamification was used to guide the general design [38]. An introductory video similar to that of recreational escape rooms was recorded to explain the case before entering the room (Appendix A). Within 60 min, the participants had to be able to carry out the initial evaluation of the patient’s severity using the paediatric evaluation triangle, monitor the patient, initiate oxygen therapy, place a peripheral line, extract blood samples and cultures, and administer treatment: a bolus of saline serum and antibiotic therapy with cefotaxime at the correct doses. All this was carried out step by step and in this order with the clues that were sequentially obtained (Figure 1 and Figure 2). During the escape room, the game masters followed the case from outside the room at all times in order to fully assess leadership and teamwork. We repeated the escape room 11 times, each time with a new team of 4–5 residents. Each group created was composed of a mixture of doctors and nurses in different years of training, which makes our study different from those previously reported in the literature, where different types of students were not mixed.

Just before the escape room, a prior knowledge test based on the international guidelines for the management of severe sepsis and septic shock was carried out [39] (Table 1), which was repeated right at the end of the escape room and three months later to compare not only the knowledge acquired at the moment but also how it was assimilated and retained over time.

At the end of the escape room, all participants underwent a formal debriefing on the diagnosis and initial management of septic shock in paediatrics and completed an anonymous post-study survey with the objective of testing the first two levels of Kirkpatrick’s levels of training evaluation [40]. These measurements are useful in training, especially to internally measure the quality of the programs designed and delivered.

Kirkpatrick level 1 aims to measure participant satisfaction with the training and find out to what extent they find the training favorable, engaging, and relevant to their jobs. Level 2 analyzes the degree to which participants acquire the intended knowledge, skills, attitude, confidence, and commitment, based on their participation in the learning event.

The sociodemographic variables sex, age, medical or nursing resident, and year of residence were collected. We also recorded the number of recreational escape rooms that the participants had previously completed, the number of clues they asked for, and whether they completed the mission on time.

### 2.1. Statistical Analysis

Qualitative variables were expressed as absolute and relative values, whereas quantitative variables were defined using mean ± standard deviation or median and interquartile range (IQR), depending on whether they were normally distributed. The comparison of quantitative variables was performed with the Wilcoxon test. The significance level was set at 0.05, and the statistical analysis was carried out with SPSS ^®^ 21.0 software (SPSS Inc., Chicago, IL, USA).

### 2.2. Ethics Statement

This study was approved by the institutional Clinical Research Ethics Committee in compliance with the Declaration of Helsinki (last update, Fortaleza, 2013).

## 3. Results

A total of 48 participants were recruited: 10 (20.8%) first-year medical residents, 11 (22.9%) second-year medical residents, 9 (18.8%) third-year medical residents, 11 (22.9%) fourth-year medical residents, 4 (8.3%) first-year nursing residents, and 3 (6.3%) second-year nursing residents. All residents are in the paediatric specialty. The mean age was 26.2 ± 2.03 years and 38 (79.2%) were women.

As for the participants’ prior experience in recreational escape rooms, the distribution is as follows: 32 (66.7%) participants did less than 5, 14 (29.1%) did from 5 to 15, 1 (2.1%) did from 15 to 30, and 1 (2.1%) participated in more than 30 escape rooms.

The mean score for the pre-escape room exam was 7.85/9 (SD 1.65), that for the post-escape room exam was 8.75/9 (SD 0.53), and three months later, this was 8.30/9 (SD 0.94). The analysis according to the type of resident is shown in Table 2 and Figure 3.

Of the participants, 18.8% of them, corresponding to two teams (nine people), did not manage to leave before the established 60 min. The mean number of clues the participants asked for or were given by the game master was 10.5 (SD 2.7) overall.

Finally, the results of the satisfaction survey are shown in Table 3 and Figure 4.

## 4. Discussion

This manuscript presented the results of a new way of learning for paediatric medical residents and paediatric nursing residents: carrying out an escape room based on a septic patient.

Learning games are used for various reasons, such as being able to immerse the player; motivating the student; enhancing teamwork, communication, and leadership; and being able to offer a safe environment in which errors have no consequences except within the game itself [3,4]. The difficulty lies in how to make the scenario mimic real life and encouraging the involvement of the participants.

The construction of the escape room was performed according to an established order: selection of the problem to be worked on, organization of content that the escape room aimed to deliver, selection of activities and fundamental questions, and selection of means and evaluation of the learning processes. Nicholson’s RECIPE mnemonic (reflection, engagement, choice, information, play, exposition) for meaningful gamification was used to guide the general design [38]. Sepsis was considered as a good topic because, due to its frequency and severity, it is important that all health personnel know the steps to be followed when facing this medical problem.

Due to the novelty of the escape room concept, there is a lack of research that reports on and rigorously examines the potential educational use of these rooms. However, as of late, several articles are being published in a multitude of fields other than medicine, such as pharmacy, physical therapy, chemistry, computer networks, cryptography, mathematics, and programming [2,9,10,11,12,13,14,15,16,17,18,19,20,21,22,23,24,25,26,27,28,29,30,31,32,33]. There are also initiatives targeting pre-university students in different fields [34,35,36,37]. There have even been escape rooms about COVID-19 [41,42], and nursing also has several publications on these experiences [16,17,18,19]. In view of the results of our study and the few articles published in the field of paediatrics, it seems to be a subject with a large potential for exploitation.

An important challenge to take into account is adjusting the level of difficulty, both in the games and in the mastery of the conceptual material for the knowledge area in question [8,43]. Complexity plays a crucial role. If it is too simple, it will bore the students, but if it is too difficult, it can produce frustration or anxiety. This delicate balance consists of keeping the player in a state of total immersion in which he or she finds the challenges neither too difficult nor too simple. That is why in our case, the participants were given the freedom to ask for as many clues as they considered necessary, making for a very fluid game. In the cases where the game masters detected stagnation in the progress of the escape room, they also gave some clues, which were recorded.

According to the bibliography consulted, there are no works prior to 2017 that seriously report on the success rate of the use of educational escape rooms, that is, the percentage of students who solve the room before time runs out [44]. This type of data is essential to assess the difficulty of the activity and modulate it in future runs. In our study, 81.2% of the residents were able to successfully exit the room within the required 60 min. We gave the participants all the time they needed to get to the end in order to completely achieve the objectives, although this was one of the variables that was recorded.

To evaluate this as a learning system, a sepsis test was performed based on international guidelines for the management of severe sepsis and septic shock [39], both pre-escape room and after this activity. Furthermore, it was repeated 3 months later in order to assess the degree of knowledge acquired and retained over time. The results show an improvement in the score after carrying out the activity that persists over time, to a somewhat lesser degree. However, the initial result was already very high, suggesting that the test used was easy for all participants. It would be interesting to periodically repeat the session in order to consolidate the learning.

Finally, as for the perception of the participants regarding educational escape rooms, both the bibliography and our own data mention that the residents enjoyed participating and that, at the same time, they considered the experience to be valuable for learning [12,21,45,46]. Participants prefer escape rooms over conventional teaching experiences [23,25] and would recommend it to their colleagues. Despite this, the similarity of acting and communication to that of real life were the worst valued items. This means that a greater effort should be undertaken in the transformation of the scenario, making it more realistic.

The main limitation of the study is the novelty of this type of learning technique, both for students and for the game masters tasked with designing these learning sessions. The different levels of previous experience in recreational escape rooms that participants had may entail a bias. On the other hand, the sample size is still too small to be able to extrapolate these data to the general population. The fact that participants are working with a mannequin in a simulation room can prevent them from performing their usual way of working with patients on a daily basis, despite being previously reminded about the importance of getting into the role. Finally, to improve the test in the future, access to models with racial diversity would be valuable and reduce the inequity that can occur if skin tone is not medically taken into consideration. For example, cyanosis or the sepsis rash can be difficult to detect or look different on a darker skin when compared with a lighter skin. This training gap can lead to inequities in care.

## 5. Conclusions

Gamification as a new form of learning for paediatric medical residents and nursing residents seems to have a positive educational impact that was sustained over time. In addition, the level of satisfaction of the participants was very good, so this method seems to have great prospects for the future.

## Figures and Tables

**Figure 1 children-09-01503-f001:**
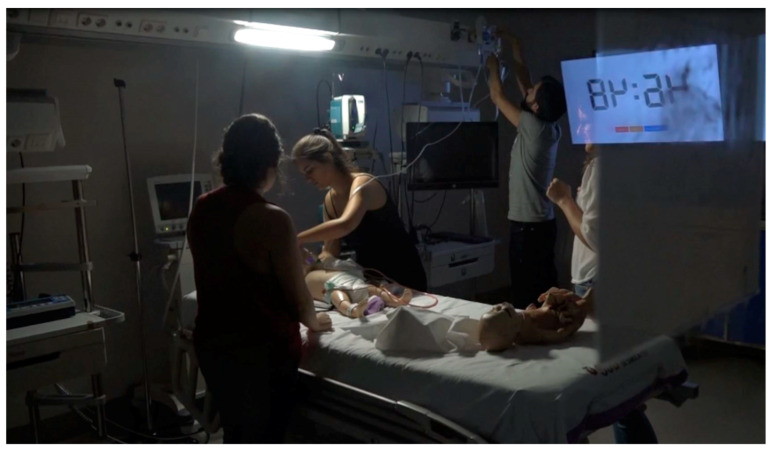
Photographs during the escape room.

**Figure 2 children-09-01503-f002:**
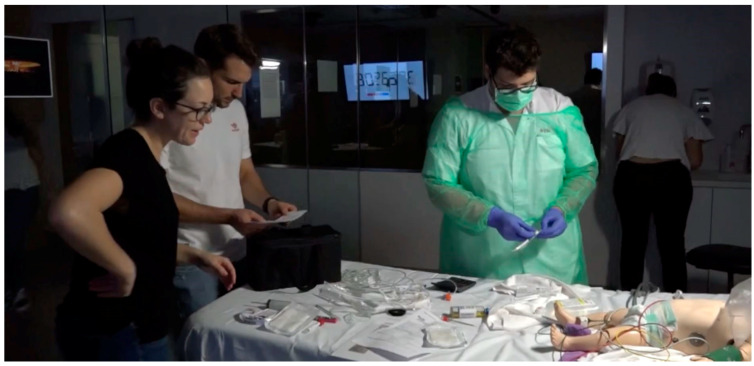
Photographs during the escape room.

**Figure 3 children-09-01503-f003:**
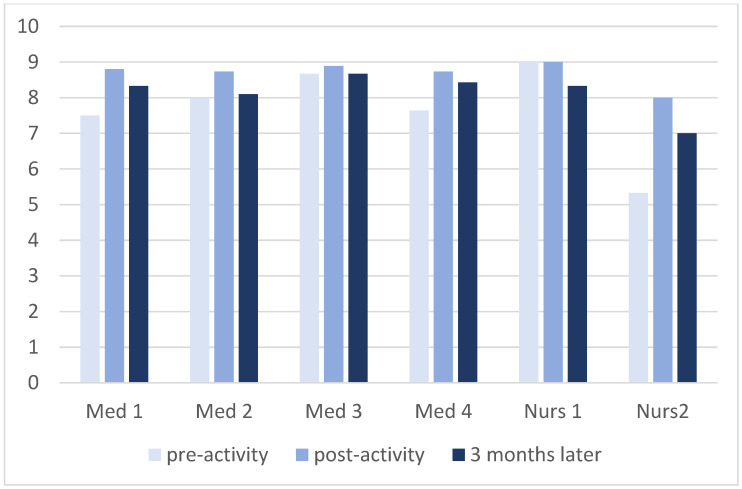
Exam score according to the type of resident. Maximum score: 9.

**Figure 4 children-09-01503-f004:**
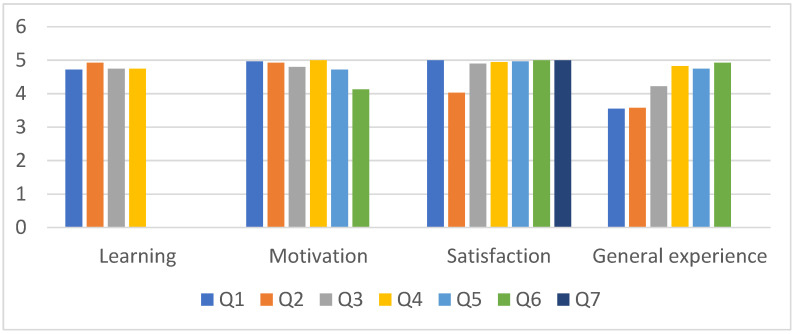
Graphic representation of satisfaction survey. Q = question (see Table 3); Maximum score: 5.

**Table 1 children-09-01503-t001:** Knowledge test.

1—Is sepsis a potentially serious condition?	**A**—Yes **B**—No
2—What is the immediate initial assessment to perform in a septic patient?	**A**—ABCDE **B**—Monitoring **C**—Pediatric Assessment Triangle **D**—The clinical eye of the doctor
3—What monitoring is necessary?	**A**—Heart rate, temperature, hemoglobin saturation, capnography, and invasive blood pressure **B**—Heart rate, respiratory rate, temperature, capnography, and mixed venous saturation **C**—Respiratory rate, heart rate, temperature, hemoglobin saturation, and noninvasive blood pressure **D**—Heart rate, invasive blood pressure, hemoglobin saturation, and respiratory rate
4—How long does it take to stabilize?	**A**—1 h **B**—24 h **C**—There is no stabilization time **D**—2 h
5—What type of fluid is initially used to stabilize sepsis?	**A**—5% glucose serum **B**—Glucosaline serum 1/3 **C**—Saline serum **D**—Blood products
6—What dose of fluid is used in the first administration?	**A**—Basal needs **B**—20 mL/kg **C**—10 mL/kg **D**—High infusion rate; the volume is irrelevant
7—What antibiotic is empirically applied in sepsis?	**A**—Meropenem **B**—Amoxicillin–clavulanic acid **C**—Cefotaxime **D**—Piperacillin–tazobactam
8—Is oxygen necessary if the hemoglobin saturation is correct?	**A**—Yes **B**—No
9—If yes, what oxygen device should be applied?	**A**—Nasal cannula 0.5 L/min **B**—Reservoir mask 15 L/min **C**—50% venturi mask **D**—100% venturi mask

**Table 2 children-09-01503-t002:** Evaluations according to the type of resident (maximum score: 9).

Type of Resident (n)	Pre-Escape Room Exam Mean (SD) (A)	Post-Escape Room Exam Mean (SD) (B)	Exam 3 Months Later Mean (SD) (C)	*p* Value (A–B)	*p* Value (B–C)	*p* Value (A–C)
Total (48)	7.85 (1.65)	8.75 (0.53)	8.3 (0.94)	0.000	0.001	0.053
First-year medical residents (10)	7.50 (1.51)	8.80 (0.42)	8.33 (1.12)	0.013	0.104	0.040
Second-year medical residents (11)	8 (1.48)	8.73 (0.47)	8.10 (0.74)	0.070	0.051	0.823
Third-year medical residents (9)	8.67 (0.71)	8.89 (0.33)	8.67 (0.5)	0.169	0.169	1
Fourth-year medical residents (11)	7.64 (1.75)	8.73 (0.65)	8.43 (0.79)	0.052	0.457	0.143
First-year nursing residents (4)	9 (0)	9 (0)	8.33 (0.58)	1	0.184	0.184
Second-year nursing residents (3)	5.33 (3.06)	8 (1)	7 (2.83)	0.251	0.795	0.590

**Table 3 children-09-01503-t003:** Results of satisfaction survey (1 = strongly disagree, 5 = strongly agree).

Part 1: Learning	Mean (SD)
Q1: The game helped me learn about sepsis	4.72 (0.45)
Q2: I find it useful as a training method	4.93 (0.27)
Q3: I find it more useful than purely theoretical classes	4.75 (0.49)
Q4: I have applied my knowledge during the game	4.75 (0.54)
Part 2: Motivation	Mean (SD)
Q1: The topic is interesting to me	4.97 (0.16)
Q2: The format helped me learn	4.93 (0.27)
Q3: The game motivated me to expand my knowledge about sepsis	4.80 (0.41)
Q4: I felt motivated to carry out the activity	5 (0)
Q5: It motivated me in the use of communication skills	4.72 (0.60)
Q6: It motivated me in the use of leadership skills	4.13 (0.91)
Part 3: Satisfaction	Mean (SD)
Q1: I enjoyed the activity	5 (0)
Q2: The format is not stressful	4.03 (0.92)
Q3: The format is appropriate	4.90 (0.30)
Q4: There should be more training based on this methodology in my profession	4.95 (0.22)
Q5: The activity met my expectations	4.97 (0.16)
Q6: I would recommend it to other residents	5 (0)
Q7: General degree of satisfaction	5 (0)
Part 4: General experience	Mean (SD)
Q1: My performance in the game was similar to that of a real case	3.55 (0.96)
Q2: My communication in the game was similar to that of a real case	3.58 (0.90)
Q3: Teamwork was similar to that of a real case	4.22 (0.66)
Q4: I think it can help me in a real case	4.83 (0.38)
Q5: I have applied my knowledge during the game	4.75 (0.49)
Q6: What I have learned in the game will help me in real life	4.93 (0.27)

Q = question.

## Data Availability

The data was stored in an encrypted database in our hospital.

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
