# Peer review of "New Resident Training Strategy Based on Gamification Techniques: An Escape Room on Sepsis in Children"

_children, 2022, doi:10.3390/children9101503_

Round 1
Reviewer 1 Report
Many thanks for providing me with the opportunity to review this manuscript. It presents a game-based escape room activity for aiding nursing students' knowledge of sepsis.
The manuscript has no changes from me as I think it reads very well and is certainly a topic that will prove interesting to readers. It is very easy to understand and does a good job of explaining the data as well as setting the results in the context of the present literature. Escape rooms are something that are not entirely novel in an educational setting but there are still many colleagues who would benefit from another fine example of the use of gamification in an education setting.
My only comment for a change would be in your results, specifically Table 2 and Table 3. It would be nice to see these represented graphically as bar charts with associated error bars and significance values shown. This is a minor comment and is likely to be a personal view from myself but if it was possible to create graphical representations I feel it would aid the reader in making comparisons between the different data sets.
I was also unable to view the video content as I was only able to download and review the manuscript. Viewing these videos would in no way change my recommendation for this work to be accepted but I wanted to highlight it here - I have also highlighted this to the Editors as it would be useful to have a way of reviewers seeing digital content such as videos, animations, etc.
I commend you on an excellently written study and wish you all the very best in your future research endeavours.
Author Response
Reviewer 1:
Many thanks for providing me with the opportunity to review this manuscript. It presents a game-based escape room activity for aiding nursing students' knowledge of sepsis.
The manuscript has no changes from me as I think it reads very well and is certainly a topic that will prove interesting to readers. It is very easy to understand and does a good job of explaining the data as well as setting the results in the context of the present literature. Escape rooms are something that are not entirely novel in an educational setting but there are still many colleagues who would benefit from another fine example of the use of gamification in an education setting.
My only comment for a change would be in your results, specifically Table 2 and Table 3. It would be nice to see these represented graphically as bar charts with associated error bars and significance values shown. This is a minor comment and is likely to be a personal view from myself but if it was possible to create graphical representations I feel it would aid the reader in making comparisons between the different data sets.
I was also unable to view the video content as I was only able to download and review the manuscript. Viewing these videos would in no way change my recommendation for this work to be accepted but I wanted to highlight it here - I have also highlighted this to the Editors as it would be useful to have a way of reviewers seeing digital content such as videos, animations, etc.
I commend you on an excellently written study and wish you all the very best in your future research endeavours.
Thank you very much for all your comments. The truth is that reviewers like you encourage new studies. We are extremely happy and satisfied that this work is of interest to you. We fully agree with your comment that some graphics would improve the understanding of Tables 2 and 3. That is why we have added some graphics to help the understanding of these tables.
Reviewer 2 Report
Review of “2 Title: New resident training strategy based on gamification techniques: An escape room 3 on sepsis” Children
Page 2 line 31 “The pediatric and pediatric nursing 31 residents were enrolled” I am not sure why pediatric is in this sentence twice? Should this say pediatric medical residents and nursing residents?
Line 35-36 “The mean score for 35 the pre-escape room exam was 7.85/9 (SD 1.65), that for the post-escape room exam 36 was 8.75/9 (SD 0.53), and for the exam three months later it was 8.30/9 (SD 0.94).” This seems that the pre-exercise exam scores are very high (87.22%) and went up to 97.22% which is good, but is this activity necessary if the scores are already that high? IT does appear that the scores remained higher than the pre-tests even at 3 months, but do you have a comparison of students who did not do the activity at 3 months?
One interesting thing that I noted when looking at your data that was not expanded upon was that the first year med students seemed to gain the most from this activity.
Also, what is really novel from what I can tell about this study is the mixing of all levels of students including both MD and Nursing students, this needs to be made more clear, as there are lots of papers on educational escape rooms as is demonstrated by your bibliography. You need to make more clear how your paper is different than the rest. This should go into your abstract, as should the statistical analysis.
line 154-155 “154 This manuscript presents the results of a new way of learning for residents: carrying out 155 an escape room based on a septic patient.” do none of the other publications include residents?
Lines 169-176 this paragraph needs to be reworked it states “Due to the novelty of the escape room concept, there is a lack of research that reports on 170 and rigorously examines the potential educational use of these rooms. However, as of 171 late several articles are being published in a multitude of fields other than medicine and 172 nursing, such as pharmacy, physical therapy, chemistry, computer networks, cryptography, mathematics, and programming2,9-33 173 . There are also initiatives targeting pre-university students in different fields34-37 174 . There have even been escape rooms about COVID-1941,42 175 . In view of the results of our study, it seems to be a subject with a large 176 potential for exploitation” However there are at least 4 references that are about escape rooms in Nursing. (see below) So these statements are not quite accurate and need to be reworked.
16. Adams V, Burger S, Crawford K, Setter R. Can you Escape? Creating an Escape 316 Room to facilitate active learning. J Nurses Prof Dev. 2018; 34(2):E1-E5. 317 318
17. Brown N, Darby W, Coronel H. An Escape Room as a simulation teaching 319 strategy. Clinical simulation in nursing. 2019; 30:1-6. 320 321
18. Castro TC, Gonçalves LS. The use of gamification to teach in the nursing field. 322 Rev Bras Enferm. 2018; 71(3):1038-1045. 323 324
19. Morrell BLM, Eukel HN. Escape the generational gap: a cardiovascular escape 325 room for nursing education. J Nurs Educ. 2020; 59(2):111-115.
Line 205-206 “Despite this, the similarity 206 of acting and communication to that of real life were the worst valued items.” this is an awkward sentence.
Line 209-210 “, both 210 for students and for the game masters entasked with designing these learning sessions” It should be …game masters TASKED with…
Line 223 “Gamification as a new form of learning for pediatric and nursing residents seems to” This is not clear to me does this mean pediatric MD residents and nursing residents? Clarify
I would discuss the differences you see in improvements in the different groups of residents who participated, although the numbers are small, so statistical significance is hard to see, you do see some interesting differences. The 1st year MD residents definitely improved the most, and oddly then the 4th year MR. This warrants some discussion.
Author Response
Reviewer 2:
Review of “Title: New resident training strategy based on gamification techniques: An escape room on sepsis” Children. Thank you for your comment. We have modified the title according to your suggestion.
Page 2 line 31 “The pediatric and pediatric nursing residents were enrolled” I am not sure why pediatric is in this sentence twice? Should this say pediatric medical residents and nursing residents? This is exactly what we wanted to write. Sentence changed for better understanding.
Line 35-36 “The mean score for the pre-escape room exam was 7.85/9 (SD 1.65), that for the post-escape room exam was 8.75/9 (SD 0.53), and for the exam three months later it was 8.30/9 (SD 0.94).” This seems that the pre-exercise exam scores are very high (87.22%) and went up to 97.22% which is good, but is this activity necessary if the scores are already that high? IT does appear that the scores remained higher than the pre-tests even at 3 months, but do you have a comparison of students who did not do the activity at 3 months? Thanks for your question. It is true that the initial exam score was very high. Being the first time that we carried out a project of this type, we made the exam too easy. We saw this a posteriori and we commented on it in the discussion to improve in future projects. The important fact, as you mentioned, is that the score increases and remains high after 3 months of the escape room. Results were not compared with residents who did not do this activity as this was not an objective of this study.
One interesting thing that I noted when looking at your data that was not expanded upon was that the first year med students seemed to gain the most from this activity. Certainly, the first-year residents present a greater use of the activity since they do not have the experience or the knowledge of the less novice residents. That is why we think that this training strategy can have good results for learning.
Also, what is really novel from what I can tell about this study is the mixing of all levels of students including both MD and Nursing students, this needs to be made more clear, as there are lots of papers on educational escape rooms as is demonstrated by your bibliography. You need to make more clear how your paper is different than the rest. This should go into your abstract, as should the statistical analysis. You have all the reason. The mixture of different types of student makes our study different, as well as its separate statistical analysis. We have tried to make it clear in the manuscript.
line 154-155 “This manuscript presents the results of a new way of learning for residents: carrying out an escape room based on a septic patient.” do none of the other publications include residents? Thanks for your question. To our knowledge, our study is the first to include pediatric medical residents and nursing residents. There is some other study that includes residents, but from other specialties. We have clarified this point in the text.
Lines 169-176 this paragraph needs to be reworked it states “Due to the novelty of the escape room concept, there is a lack of research that reports on and rigorously examines the potential educational use of these rooms. However, as of late several articles are being published in a multitude of fields other than medicine and nursing, such as pharmacy, physical therapy, chemistry, computer networks, cryptography, mathematics, and programming. There are also initiatives targeting pre-university students in different fields. There have even been escape rooms about COVID-19. In view of the results of our study, it seems to be a subject with a large potential for exploitation” However there are at least 4 references that are about escape rooms in Nursing. (see below) So these statements are not quite accurate and need to be reworked.
- Adams V, Burger S, Crawford K, Setter R. Can you Escape? Creating an Escape 316 Room to facilitate active learning. J Nurses Prof Dev. 2018; 34(2):E1-E5. 317 318
- Brown N, Darby W, Coronel H. An Escape Room as a simulation teaching 319 strategy. Clinical simulation in nursing. 2019; 30:1-6. 320 321
- Castro TC, Gonçalves LS. The use of gamification to teach in the nursing field. 322 Rev Bras Enferm. 2018; 71(3):1038-1045. 323 324
- Morrell BLM, Eukel HN. Escape the generational gap: a cardiovascular escape 325 room for nursing education. J Nurs Educ. 2020; 59(2):111-115.
We are totally agreed. We have reworked this paragraph for better understanding.
Line 205-206 “Despite this, the similarity of acting and communication to that of real life were the worst valued items.” this is an awkward sentence. You are absolutely right, but these are the results of our study. The lesson to be learned is the improvement of the scenarios to resemble reality, as explained in the text. However, if you prefer to remove this sentence, we can do it without problems.
Line 209-210 “, both for students and for the game masters entasked with designing these learning sessions” It should be …game masters TASKED with… The error in the manuscript has been corrected.
Line 223 “Gamification as a new form of learning for pediatric and nursing residents seems to” This is not clear to me does this mean pediatric MD residents and nursing residents? Clarify. We have clarified this sentence for your better understanding.
I would discuss the differences you see in improvements in the different groups of residents who participated, although the numbers are small, so statistical significance is hard to see, you do see some interesting differences. The 1st year MD residents definitely improved the most, and oddly then the 4th year MR. This warrants some discussion. Thank you for your comment. Certainly the students who improved the most were the 1st year doctors and the only statistically significant one. The causes, as mentioned above, seem obvious, since they are the most novice. With regard to 4th year doctors, the number of participants, as you mentioned, is too low to draw any conclusions. And we must bear in mind that sometimes there are people or groups of them more or less predisposed to learn. However, we do not dare to draw any conclusions in this regard.